## [Transparent Peer Review file · Nature Communications]

RPA directly stimulates Mer3 helicase processivity to ensure normal crossover formation in meiosis

Corresponding Author: Dr John Weir

Version 0:

Reviewer comments:

Reviewer #1

(Remarks to the Author)

In Altmannova et al, the authors describe a novel interaction that forms between Mer3 and RPA. Mutation of this region to alanine results in the disruption of RPA binding, leading to an increase in non-crossover outcomes. The interaction studies and the in vivo analysis are well done, and the conclusions are supported by the data. The single-molecule experiments require further work, and it remains unclear whether there is an actual difference between the WT and mer3-8A mutant that is dependent on RPA. It looks like basal helicase function is different between the WT and 8A mutant. This section of the manuscript requires significant revision. I think that Nature Communications is a suitable venue for this manuscript. The study is interesting, and it will be of general interest to a broad readership. However, I found the manuscript difficult to read, and it contained numerous typos. The figures also appear slapped together and lack clear organization. There appear to be more experiments than necessary to test the specific questions in the manuscript. I think this makes it difficult for the reader to evaluate the science, which is good. There will need to be a significant amount of textual revision for clarity before I can support this manuscript for publication. Below, I have detailed some major and minor points that also need to be addressed.

Major:

Paragraph Starting with line 280: I don't think that using this type of cross-linking is a good measure of conformational dynamics. Cross-linking, by nature, is a non-equilibrium technique that can trap rare events. I think this makes it difficult to determine conformational changes. It also really doesn't affect the key findings.

The single molecule experimental setup is poorly described.

Paragraph 336: Does this mean that Mer3 is a passive or active helicase?

Figure 3F: There is a difference between WT and 8A Mer3 observed at 10 pN. This data seems to suggest that the mutant is more processive than the WT at 10 pN of force, and that there is an intrinsic difference in helicase activity with this mutant. The authors don't comment on this or explain this difference, and they should. This would mean the conclusion in Line 345 is incorrect.

In figure 4 why is the mutant protein used at 200 pM but the WT at 100 pM? Could the differences observed in activity be due to differences in the concentration of the protein? There are unmatched data points in this graph. I think this graph needs to be reconstructed so that all conditions have the same data points and the Mer3 is used at the same concentration. This is also true for 4C. The data are not convincing because of the poor presentation.

Figure 4D: Where is the 200 pM Mer-8A alone?

Discussion:

Line 557 Sequencing RPA seems unlikely given the high concentration. Its possible for Top3-Rmi1 because this is likely in lower concentration.

Minor:

Line 167: I really don't understand this paragraph. The experiment needs to be motivated better.

Line 171: ...at saturating concentrations

Line 266 space

Line 274 space

Line 327 "...should result in processive..."

Line 552 space

Reviewer #2

(Remarks to the Author)

This manuscript by Altmannova et al. investigates the interaction between a meiosis-specific helicase Mer3 and the RPA subunit Rfa1, as well as its significance in crossover formation. Using biochemical assays and AlphaFold predictions, the authors demonstrate that Mer3 directly interacts with the N-terminal OB fold of Rfa1 and identify an eight-amino acid region within Mer3 that partially mediates this interaction. Using magnetic tweezers with purified components, they show that Mer3 binding to RPA enhances its processivity. Finally, genetic analyses in *S. cerevisiae* reveal that a Mer3 mutant exhibits modestly reduced spore viability, an increase in non-crossover events, and impaired recruitment of Mer3 to DSB sites.

Overall, the manuscript is well written, and the data are of high quality that support the main conclusions. However, the interaction between Mer3 and Rfa1 is mediated through multiple interfaces, such that disrupting a single conserved interface characterized here only partially reduces binding and leads to modest defects in crossover formation in budding yeast. While the phenotypic consequences are relatively limited, the findings remain of interest and provide valuable insight into the mechanism of Mer3 function. Specific points are listed below:

1. Lines 39-40, please move "of the mer3 mutant" before "to double-strand break sites" for clarity.
2. Figure 1A: Please define the acronyms for Mer3 domains in the figure legends.
3. Figure 1B was never called in the text.
4. Line 183: It may be clearer to first describe the multiple self-crosslinks and then use these to support the >1:1:1 stoichiometry. Also, Coomassie staining in Figure 1E suggests that RPA binding to Mer3 is substoichiometric in the absence of DNA. Just out of curiosity, have you considered predicting the Mer3:RPA complex with DNA using AlphaFold3 to see if it might provide additional insights?
5. Figure legends for Supplementary Figure 7A-B are missing.
6. Supplementary Figure 7F: A schematic illustrating the strand separation assay would be helpful for readers who are unfamiliar with this method.
7. Figure 4: Authors use single-molecule assays with optical tweezers to gain more insight into the effect of RPA on Mer3 unwinding efficiency. Comparison between WT and 8A mutant, however, is done at different protein concentrations, and there is no explanation in the text or methods for this difference.
8. Line 365, it should have been Supplementary Figure 9D. In Supplementary Figure 9D, what do the colors represent? Please describe them in the legends.
9. Figures 5I and K: statistical tests are necessary to state that there are differences between WT and mutant.
10. Figure 5L was never called in the text.

Reviewer #3

(Remarks to the Author)

The study by Veronika Altmannova and colleagues identifies a direct interaction between the OB-fold of the RPA large subunit Rfa1 and a short C-terminal α -helix of Mer3 (HFM1 in humans), and shows that this interaction stimulates Mer3 helicase processivity. By integrating biochemical interaction assays, AlphaFold2-based structural modeling, single-molecule magnetic tweezers analysis, and in vivo functional assays, the authors present a coherent molecular framework for how the RPA–Mer3 interaction promotes crossover formation during meiosis. Overall, the manuscript is logically consistent and, based on the presented data, free of obvious flaws.

1. I recommend that the authors further substantiate the direct interaction between the Rfa1 N-OB domain and the Mer3 C-terminal peptide by performing quantitative biophysical measurements using isolated components:

Conduct ITC experiments with the isolated Rfa1 N-OB domain and a synthetic Mer3 C-terminal peptide to determine the binding thermodynamics and affinity. In parallel, test the Mer3 C-peptide-8A mutant to quantify the loss of binding.

In the pull-downs with full-length proteins, the 8A mutant still shows residual binding, consistent with the manuscript's proposal of a secondary, lower-affinity contact on the Mer3 helicase core. Using the minimal OB domain and short Mer3 peptides in ITC would more cleanly resolve the high-affinity site in isolation and provide orthogonal validation to the MST data reported.

2. I also suggest adding AlphaFold2 multimer predictions for the human orthologs (RPA and HFM1) focusing specifically on

the RFA1 N-OB domain and the C-terminal helix of HFM1:

This would complement the existing yeast modeling and the human pulldown/Y2H data, and help assess whether the same N-OB–helix docking mode is conserved in detail in the human complex.

Version 1:

Reviewer comments:

Reviewer #1

(Remarks to the Author)

Most of my concerns were addressed with the revision. I still have a problem with how the mutant was treated in the single molecule activity assay. However, I think the authors' approach is logical and warrants publication, so I don't want to hold up acceptance of the manuscript. I have two minor concerns and one comment.

Minor:

Figure 2D shows a random $K_d = 17.4$ associated with the sequence alignment. Line 318 by SSB binding. The word binding is missing here.

Comment:

The 8A mutant is different from the WT in the absence of RPA. This doesn't change your conclusions, but you spend a lot of energy trying to convince the reader that there is no difference in basal activity between the WT and 8A mutant. This is very frustrating and was not addressed upon my initial review. The 8A mutant is also defective in DNA binding. Therefore, the RPA binding site probably evolved later to stabilize an existing interaction that promoted helicase binding to DNA.

Reviewer #2

(Remarks to the Author)

The authors have addressed all the previous concerns.

Reviewer #3

(Remarks to the Author)

The authors' revisions have adequately addressed the concerns raised, including the additional experiments requested. No further comments or concerns remain.

Rebuttal to Reviewer Comments on Altmannova *et al.*

Reviewer #1 (Remarks to the Author):

In Altmannova *et al.*, the authors describe a novel interaction that forms between Mer3 and RPA. Mutation of this region to alanine results in the disruption of RPA binding, leading to an increase in non-crossover outcomes. The interaction studies and the *in vivo* analysis are well done, and the conclusions are supported by the data. The single-molecule experiments require further work, and it remains unclear whether there is an actual difference between the WT and mer3-8A mutant that is dependent on RPA. It looks like basal helicase function is different between the WT and 8A mutant. This section of the manuscript requires significant revision. I think that Nature Communications is a suitable venue for this manuscript. The study is interesting, and it will be of general interest to a broad readership. However, I found the manuscript difficult to read, and it contained numerous typos. The figures also appear slapped together and lack clear organization. There appear to be more experiments than necessary to test the specific questions in the manuscript. I think this makes it difficult for the reader to evaluate the science, which is good. There will need to be a significant amount of textual revision for clarity before I can support this manuscript for publication. Below, I have detailed some major and minor points that also need to be addressed.

We thank the reviewer for their comments, and that they find the study interesting, of interest to a broad audience, and suitable for Nature Communications.

As requested we have revised the manuscript, and paid particular attention to readability and to the organisation of the figures. We have endeavoured to produce a more streamlined set of main figures, and made greater use of supplementary figures for those readers who would like to go deeper. In particular we have revised the presentation of the single molecule data, which is described in detail below.

Major:

Paragraph Starting with line 280: I don't think that using this type of cross-linking is a good measure of conformational dynamics. Cross-linking, by nature, is a non-equilibrium technique that can trap rare events. I think this makes it difficult to determine conformational changes. It also really doesn't affect the key findings.

We have removed this interpretation of the cross-linking data from the manuscript.

The single molecule experimental setup is poorly described.

We apologise for this. We have expanded the description of the single-molecule experimental setup in the main text and in the Supplementary Information to provide a clearer and more detailed explanation of the experimental design, data acquisition and analysis.

Paragraph 336: Does this mean that Mer3 is a passive or active helicase?

Our data suggests that Mer3 functions as an active helicase for two main reasons:

- i) its unwinding velocity is force-independent, and
- ii) the unwinding and translocation rates are similar.

The point (i) is shown in Figure 3E for Mer3-WT and Mer3-8A.

To answer the point (ii) the translocation rate was calculated by performing cycles of opening/closing of the hairpin as it has been previously shown (Bagchi D., et al. Single molecule kinetics uncover roles for *E. coli* RecQ DNA helicase domains and interaction with SSB. *Nucleic Acid Research*. 2018; 8500-8515. doi: 10.1093/nar/gky647; Klaue D., et al. Fork sensing and strand switching control antagonistic activities of RecQ helicases. *Nature Communications*. 2013; 4:2024. doi: 10.1038/ncomms3024).

Briefly, during an unwinding event, the hairpin is either unzipped ($F > 15$ pN) or mechanically opened for a time interval Δt and then reziped once the force is reduced ($F < 15$ pN). During Δt , the helicase can translocate along the ssDNA. When the force is reduced, the hairpin refolds, producing a significant change in extension. This change indicates that the enzyme has moved a distance Δz along ssDNA during Δt . The helicase activity can be measured just before and just after the hairpin opening, and its translocation on ssDNA can be inferred from the continuity of the extension trace. After the hairpin refolds, the enzyme switches strands and starts unwinding the hairpin again.

Figure R1. Translocation of Mer3-WT (green) and Mer3-8A (orange) along single-stranded DNA. We have added a new figure (Supplementary Figure 8C) to the Supplementary Information showing that the unwinding and translocation rates are comparable for both Mer3-WT and Mer3-8A under our experimental conditions. However, we point that a detailed analysis of translocation lies outside the scope of our current study and will be explored in future work.

Figure 3F: There is a difference between WT and 8A Mer3 observed at 10 pN. This data seems to suggest that the mutant is more processive than the WT at 10 pN of force, and that there is an intrinsic difference in helicase activity with this mutant. The authors don't comment on this or explain this difference, and they should. This would mean the conclusion in Line 345 is incorrect.

We thank the reviewer for raising this comment. To address this point, we performed thorough additional experiments at 10 pN for both Mer3-8A (increasing the number of unwinding events from 8 to 23) and Mer3-WT (from 34 to 53) to improve the statistical robustness and determine whether the apparent difference between the two proteins was genuine or simply due to the previously limited dataset at this particular force. Additional measurements at higher forces have also been included to strengthen the statistical analysis, and the updated data are now shown in the revised Figure 3E and 3F. The characteristic traces of Mer3-WT (Fig. 3C) and Mer3-8A (Fig. 3D) have been replaced with new examples recorded at the same constant force for better understanding by the reader. Additional examples at different applied forces are now provided in the Supplementary Information (Supplementary Figure 8D).

Figure R2. Single-molecule analysis of Mer3 helicase activity. C, D) Representative single-molecule unwinding traces measured at (C) 100 pM Mer3-WT and (D) 200 pM Mer3-8A under 14 pN applied force. Arrows indicate Mer3 dissociation following complete hairpin opening, after which the hairpin rapidly closes. E) Unwinding rates for Mer3-WT (green) and Mer3-8A (orange) at forces ranging from 14 to 5.5 pN. Data were fitted with a Gaussian

function. A Mann-Whitney test showed no significant difference between Mer3-WT and Mer3-8A at any force ($P > 0.05$), with both variants exhibiting a force-independent unwinding rate of ~ 21 bp/s. F) Length of DNA unwound by Mer3-WT (green) and Mer3-8A (orange) over the same force range. A Mann-Whitney test showed no significant difference between variants ($P > 0.05$) except at 10 pN ($P = 0.01553$). At forces > 10 pN, both proteins displayed processivities of ~ 1200 bp, which dropped substantially at lower forces. Box plots in E) and F) indicate the mean, median, 25th and 75th percentiles of the distributions, and the whiskers denote the standard deviation.

To better illustrate the behaviour of both proteins at 10 pN, we plotted the density distributions of unwinding lengths for Mer3-WT and Mer3-8A proteins (Figure R3. A):

Figure R3. Mer3 shows a bi-modal processivity distribution at 10 pN. A) Density distributions of unwinding lengths for Mer3-WT (green) and Mer3-8A (orange) proteins at 10 pN. B) Sigmoidal fits of median processivities at different applied forces for Mer3-WT (green) and Mer3-8A (orange) proteins.

These distributions reveal two distinct populations corresponding to short and long DNA-unwinding events, suggesting that 10 pN applied force likely represents an inflection force-point where processivity begins to decline since at forces below 10 pN the processivity sharply drops for both proteins. At 10 pN, both full (long-length) and partial (short-length) DNA-unwinding events coexist for Mer3-8A (orange) and Mer3-WT (green), explaining the apparent difference in processivity observed at 10 pN. To quantitatively determine the inflection point separating high and low processivity regimes, we plotted the median and median absolute deviation of processivity as a function of applied force and fitted the data with a sigmoidal function (Figure R3 B):

$$y = \frac{A_2 + (A_1 - A_2)}{1 + e^{-(x-x_0)/dx}}$$

where x_0 represents the inflection point and dx is the slope, indicating the sharpness of the transition.

The fits yielded $x_0 = 10.27 \pm 0.18$ pN and $dx = 0.29 \pm 0.13$ bp/pN for Mer3-WT (green line), and $x_0 = 9.58 \pm 0.44$ pN and $dx = 0.38 \pm 0.26$ bp/pN for Mer3-8A (orange line). These results indicate

that the transition is steep ($dx < 0.5$) for both proteins, with the transition for Mer3-8A occurring at a slightly lower force (9.58 pN). This shift explains the somewhat higher fraction of long unwinding events observed for Mer3-8A compared to for Mer3-WT (see density distributions graph above), suggesting a subtle mechanistic difference between the two proteins.

We have added an explanation of this effect observed at 10 pN in the revised main manuscript, as well as a new figure in the Supplementary Information (Supplementary Figure 9).

In figure 4 why is the mutant protein used at 200 pM but the WT at 100 pM? Could the differences observed in activity be due to differences in the concentration of the protein?

We thank the reviewer for this observation. At 100 pM, Mer3-8A protein induced less unwinding events, which limited the statistical significance of the analysis. This likely reflects a lower ssDNA binding affinity of Mer3-8A compared to Mer3-WT under these very low concentration conditions. We have added a comment on this possibility in the revised text. To overcome the limited number of events, we increased the Mer3-8A concentration to 200 pM, while still remaining well within the single-molecule regime, as confirmed by the characteristic behavior of individual unwinding traces.

To further clarify this point, we have added representative traces to the Supplementary Information (Supplementary Figure 8B) showing individual Mer3-8A (orange) and Mer3-WT (green) activities recorded at both 100 pM and 200 pM under 13 pN force. The highly similar behaviors observed at the two concentrations confirm that, at least within this range, increasing protein concentration does not alter the single-molecule nature of the helicase measurements. We also note that, although the reviewer referred to Figure 4, the concentration difference originally arose in Figure 3 (in the absence of RPA) in the submitted manuscript. To maintain experimental consistency, we used the same concentrations in Figure 4 (with RPA).

Figure R4. Representative single-molecule activity traces measured at 100 pM (left) and at 200 pM (right) for Mer3-WT (green) and for Mer3-8A (orange) at 13 pN force. Importantly, if Mer3 concentration had a significant influence on activity in the presence of RPA, a higher concentration (200 pM) would be expected to enhance unwinding activity. However, this was not observed (see Figure R5 D). This supports the interpretation that the differences observed in the experiments with RPA arise from intrinsic properties of the protein-protein interactions rather than from concentration-dependent effects.

There are unmatched data points in this graph. I think this graph needs to be reconstructed so that all conditions have the same data points and the Mer3 is used at the same concentration. This is also true for 4C. The data are not convincing because of the poor presentation.

We thank the reviewer for this valuable comment. We have reconstructed the graphs to include all force conditions and to ensure consistency in data presentation. In addition, we performed new measurements across multiple forces, both in the absence and in the presence of RPA, to strengthen the comparison and improve the clarity and robustness of the result. These revisions are now reflected in the updated Figures 4C and 4D of the revised manuscript:

Figure R5. Mer3-RPA interaction is required for high processivity under low tension. C) Effect of RPA on unwinding rate and D) on processivity for Mer3-WT (red downward triangles) and for Mer3-8A (yellow upward triangles) across different applied forces. For comparison, measurements for 5 nM RPA alone (gray diamonds), Mer3-WT alone (green squares), and Mer3-8A alone (orange circles) are also included. Data represent median values, and error bars correspond to the median absolute deviation.

For consistency with Figure 3, we have removed the protein concentration labels from the graph itself; however, the concentrations used are indicated in the corresponding revised figure caption in the main text.

On the other hand, it is worth noting that the unwinding rate of Mer3-8A in the presence of RPA (upward yellow triangle) at 14 pN is higher than in the absence of RPA (orange circle), as at this force RPA can also unwind the DNA duplex at high velocity (gray diamond), contributing to the observed increase in rate.

In summary, the presence of RPA enhances the processivity of Mer3-WT at low force (downward red triangles), whereas it does not increase the processivity of Mer3-8ADNA (upward yellow triangles), suggesting that Mer3-RPA interaction promotes helicase activity (Figure R5 D). Interestingly, the density distributions of unwinding lengths for Mer3-WT at 10 pN (Figure R6) showed only long-unwinding events in the presence of RPA (red) compared to the absence of RPA (green), strongly supporting the role of direct Mer3-RPA interactions during DNA unwinding.

Figure R6. Density distributions of unwinding lengths for Mer3-WT in the absence (green) and in the presence of RPA (red) at 10 pN.

Figure 4D: Where is the 200 pM Mer-8A alone?

We initially omitted the 200 pM Mer3-8A alone condition to avoid overcrowding the figure. However, we agree that including it will make the dataset more complete. We have now added this condition to the revised figure (see Figure R5).

Discussion:

Line 557 Sequencing RPA seems unlikely given the high concentration. Its possible for Top3-Rmi1 because this is likely in lower concentration.

We thank the reviewer for spotting this obvious flaw in our argument. We have amended the discussion accordingly.

Minor:

Line 167: I really don't understand this paragraph. The experiment needs to be motivated better.

We have rewritten the paragraph for a clearer explanation.

Line 171: ...at saturating concentrations

Line 266 space

Revised

Line 274 space

This paragraph has been removed.

Line 327 "...should result in processive..."

This section has been extensively re-written

Line 552 space

Corrected

Reviewer #2 (Remarks to the Author):

This manuscript by Altmannova et al. investigates the interaction between a meiosis-specific helicase Mer3 and the RPA subunit Rfa1, as well as its significance in crossover formation. Using biochemical assays and AlphaFold predictions, the authors demonstrate that Mer3 directly interacts with the N-terminal OB fold of Rfa1 and identify an eight-amino acid region within Mer3 that partially mediates this interaction. Using magnetic tweezers with purified components, they show that Mer3 binding to RPA enhances its processivity. Finally, genetic analyses in *S. cerevisiae* reveal that a Mer3 mutant exhibits modestly reduced spore viability, an increase in non-crossover events, and impaired recruitment of Mer3 to DSB sites.

Overall, the manuscript is well written, and the data are of high quality that support the main conclusions. However, the interaction between Mer3 and Rfa1 is mediated through multiple interfaces, such that disrupting a single conserved interface characterized here only partially reduces binding and leads to modest defects in crossover formation in budding yeast. While the phenotypic consequences are relatively limited, the findings remain of interest and provide valuable insight into the mechanism of Mer3 function. Specific points are listed below:

We appreciate the reviewer's support and interest in our work. We have addressed their specific points below.

1. Lines 39-40, please move "of the mer3 mutant" before "to double-strand break sites" for clarity.

Corrected

2. Figure 1A: Please define the acronyms for Mer3 domains in the figure legends.

Added

3. Figure 1B was never called in the text.

Added.

4. Line 183: It may be clearer to first describe the multiple self-crosslinks and then use these to support the >1:1:1:1 stoichiometry.

Amended.

Also, Coomassie staining in Figure 1E suggests that RPA binding to Mer3 is substoichiometric in the absence of DNA. Just out of curiosity, have you considered predicting the Mer3:RPA complex with DNA using AlphaFold3 to see if it might provide additional insights?

We have tried with different lengths and types of DNA substrate in AF3. We have also tried increasing the stoichiometry of the complex in the inputs to 2:2:2:2, but none of the predictions were convincing.

5. Figure legends for Supplementary Figure 7A-B are missing.

Added.

6. Supplementary Figure 7F: A schematic illustrating the strand separation assay would be helpful for readers who are unfamiliar with this method.

This is now Supplementary Figure 7G, and we have added a schematic.

7. Figure 4: Authors use single-molecule assays with optical tweezers to gain more insight into the effect of RPA on Mer3 unwinding efficiency. Comparison between WT and 8A mutant, however, is done at different protein concentrations, and there is no explanation in the text or methods for this difference.

We have commented on this above, as the same issue was raised by reviewer 1.

At 100 pM, Mer3-8A protein induced less unwinding events, which limited the statistical significance of the analysis. This likely reflects a lower ssDNA binding affinity of Mer3-8A compared to Mer3-WT under these very low concentration conditions. We have added a comment on this possibility in the revised text. To overcome the limited number of events, we increased the Mer3-8A concentration to 200 pM, while still remaining well within the single-molecule regime, as confirmed by the characteristic behavior of individual unwinding traces.

To further clarify this point, we have added representative traces to the Supplementary Information (Supplementary Figure 9B) showing individual Mer3-8A (orange) and Mer3-WT (green) activities recorded at both 100 pM and 200 pM under 13 pN force. The highly similar behaviors observed at the two concentrations confirm that, at least within this range, increasing protein concentration does not alter the single-molecule nature of the helicase measurements. We also note that, although the reviewer referred to Figure 4, the concentration difference originally arose in Figure 3 (in the absence of RPA) in the submitted manuscript. To maintain experimental consistency, we used the same concentrations in Figure 4 (with RPA).

8. Line 365, it should have been Supplementary Figure 9D. In Supplementary Figure 9D, what do the colors represent? Please describe them in the legends.

This is now Supplementary figure 8, and we have added the descriptions in the legends, but have also added a key in the figure.

9. Figures 5I and K: statistical tests are necessary to state that there are differences between WT and mutant.

We appreciate the reviewer's interest in statistical analysis. However:

1) with n=2 replicates, formal statistical testing would not be meaningful. With only one degree of freedom, we cannot reliably estimate variance - any calculated standard deviation simply reflects the single difference between two measurements rather than sampling from a true distribution. We therefore present our duplicate measurements as means with the error bars corresponding to the data range (not SD), allowing readers to assess the consistency of observations directly.

2) This approach is standard in the field; see for example

Joo et al. 2024: <https://academic.oup.com/nar/article/52/7/3794/7606258>

Medhi et al. 2016: <https://elifesciences.org/articles/19669>

10. Figure 5L was never called in the text.

We have added a reference to Figure 5L.

Reviewer #3 (Remarks to the Author):

The study by Veronika Altmannova and colleagues identifies a direct interaction between the OB-fold of the RPA large subunit Rfa1 and a short C-terminal α -helix of Mer3 (HFM1 in humans), and shows that this interaction stimulates Mer3 helicase processivity. By integrating biochemical interaction assays, AlphaFold2-based structural modeling, single-molecule magnetic tweezers analysis, and in vivo functional assays, the authors present a coherent molecular framework for how the RPA–Mer3 interaction promotes crossover formation during

meiosis. Overall, the manuscript is logically consistent and, based on the presented data, free of obvious flaws.

We thank the reviewer for their support.

1. I recommend that the authors further substantiate the direct interaction between the Rfa1 N-OB domain and the Mer3 C-terminal peptide by performing quantitative biophysical measurements using isolated components:

Conduct ITC experiments with the isolated Rfa1 N-OB domain and a synthetic Mer3 C-terminal peptide to determine the binding thermodynamics and affinity. In parallel, test the Mer3 C-peptide-8A mutant to quantify the loss of binding.

In the pull-downs with full-length proteins, the 8A mutant still shows residual binding, consistent with the manuscript's proposal of a secondary, lower-affinity contact on the Mer3 helicase core. Using the minimal OB domain and short Mer3 peptides in ITC would more cleanly resolve the high-affinity site in isolation and provide orthogonal validation to the MST data reported.

We have, as suggested, carried out ITC using Mer3 C-terminal peptides (WT and 8A) and the N-OB domain of Rfa1. We found that Mer3 WT peptides bound with a K_d of 17 μM , in line with a range of determined binding affinities for human RFA1 interactors. The Mer3-8A peptide showed no binding in our ITC experiments. We have included this data as a new figure (Figure 2H).

2. I also suggest adding AlphaFold2 multimer predictions for the human orthologs (RPA and HFM1) focusing specifically on the RFA1 N-OB domain and the C-terminal helix of HFM1: This would complement the existing yeast modeling and the human pulldown/Y2H data, and help assess whether the same N-OB–helix docking mode is conserved in detail in the human complex.

We carried out the AF2/AF3 predictions as suggested. Curiously, while the prediction for the mouse sequences of HFM1 and RFA1 indicated the same mode of binding (now in Supplementary Figure 6B and C), and are consistent with the homologous sequence already identified in Figure 2D, the algorithm did not make the same prediction for the human sequences. We hypothesise that this is due to a divergence in the sequence of human HFM1, where a conserved hydrophobic followed by an acidic residue are inverted (magenta circles, Figure 2D). We expect that this means that there is a weaker interaction between the HFM1 C-terminal region and the N-OB of RFA1 in humans.

REVIEWERS' COMMENTS

Reviewer #1 (Remarks to the Author):

Most of my concerns were addressed with the revision. I still have a problem with how the mutant was treated in the single molecule activity assay. However, I think the authors' approach is logical and warrants publication, so I don't want to hold up acceptance of the manuscript. I have two minor concerns and one comment.

We thank the reviewer for their swift response, and have answered in-line below.

Minor:

Figure 2D shows a random $K_d = 17.4$ associated with the sequence alignment. Line 318 by SSB binding. The word binding is missing here.

We have amended this.

Comment:

The 8A mutant is different from the WT in the absence of RPA. This doesn't change your conclusions, but you spend a lot of energy trying to convince the reader that there is no difference in basal activity between the WT and 8A mutant. This is very frustrating and was not addressed upon my initial review. The 8A mutant is also defective in DNA binding. Therefore, the RPA binding site probably evolved later to stabilize an existing interaction that promoted helicase binding to DNA.

We thank the reviewer for this insightful comment and this is a legitimate criticism. We apologise for the frustration caused. The reviewer is indeed correct that subtle differences exist between Mer3-WT and Mer3-8A in the absence of RPA. Specifically:

- 1. At 10 pN, we observe a statistically significant difference in processivity ($P = 0.01553$), though not at other forces measured.*
- 2. The force-processivity transition occurs at slightly different inflection points (10.27 pN for WT vs 9.58 pN for 8A).*
- 3. Higher concentrations of Mer3-8A were required to obtain sufficient single-molecule events.*

We have revised the text of the results and appropriate figure legends to acknowledge these differences more transparently rather than emphasizing equivalence. We have also added an extra paragraph in the discussion:

"We note that the mer3-8A mutation has subtle effects on helicase function even in the absence of RPA, including reduced efficiency of DNA loading in single-molecule assays and modest differences in force-dependent processivity. This raises the possibility that the C-terminal helix evolved dual functionality: contributing to DNA engagement while also serving as a docking site

for RPA. Under this model, RPA binding would amplify an existing, albeit weaker, helicase-DNA association. Regardless of the evolutionary trajectory, our data clearly demonstrate that direct RPA interaction is essential for the enhanced processivity observed at physiologically relevant (low-tension) conditions.”

Reviewer #2 (Remarks to the Author):

The authors have addressed all the previous concerns.

Reviewer #3 (Remarks to the Author):

The authors' revisions have adequately addressed the concerns raised, including the additional experiments requested. No further comments or concerns remain.